# Extensive Angular Sampling Enables the Sensitive Localization of Macromolecules in Electron Tomograms

**DOI:** 10.3390/ijms241713375

**Published:** 2023-08-29

**Authors:** Marten L. Chaillet, Gijs van der Schot, Ilja Gubins, Sander Roet, Remco C. Veltkamp, Friedrich Förster

**Affiliations:** 1Structural Biochemistry, Bijvoet Centre for Biomolecular Research, Utrecht University, 3584 CG Utrecht, The Netherlands; m.l.chaillet@uu.nl (M.L.C.); gijsschot@gmail.com (G.v.d.S.); s.j.s.roet@uu.nl (S.R.); 2Department of Information and Computing Sciences, Utrecht University, 3584 CC Utrecht, The Netherlands; i.gubins@uu.nl (I.G.); r.c.veltkamp@uu.nl (R.C.V.)

**Keywords:** electron cryo-tomography, particle localization and identification, template matching, GPU acceleration, volume registration

## Abstract

Cryo-electron tomography provides 3D images of macromolecules in their cellular context. To detect macromolecules in tomograms, template matching (TM) is often used, which uses 3D models that are often reliable for substantial parts of the macromolecules. However, the extent of rotational searches in particle detection has not been investigated due to computational limitations. Here, we provide a GPU implementation of TM as part of the PyTOM software package, which drastically speeds up the orientational search and allows for sampling beyond the Crowther criterion within a feasible timeframe. We quantify the improvements in sensitivity and false-discovery rate for the examples of ribosome identification and detection. Sampling at the Crowther criterion, which was effectively impossible with CPU implementations due to the extensive computation times, allows for automated extraction with high sensitivity. Consequently, we also show that an extensive angular sample renders 3D TM sensitive to the local alignment of tilt series and damage induced by focused ion beam milling. With this new release of PyTOM, we focused on integration with other software packages that support more refined subtomogram-averaging workflows. The automated classification of ribosomes by TM with appropriate angular sampling on locally corrected tomograms has a sufficiently low false-discovery rate, allowing for it to be directly used for high-resolution averaging and adequate sensitivity to reveal polysome organization.

## 1. Introduction

Cryo-electron tomography (cryo-ET) allows for the 3D imaging of macromolecules within their cellular context. In cryo-ET, a series of images from a thin, cryo-frozen biological sample is collected by rotating, most commonly along a single tilt-axis, which is referred to as a tilt-series [1]. This approach is often used for in situ imaging, increasingly in combination with focused ion beam (FIB)-milling [2], and specific cellular sub-structures can also be enriched for ex vivo imaging [3]. The projections along the different tilt angles can be merged computationally in a 3D reconstruction, a tomogram. Segmentation and particle localization methods enable the identification of different macromolecules in the tomograms, which provides information on inter-molecular interactions in the cell, which is sometimes referred to as molecular sociology [4]. While the dose sensitivity of biological cryo-fixed samples limits the resolution of the tomograms to a few nm [5], the coherent summation of the signal of copies of the same type of molecule yields significant information with a higher level of detail [6]. Reconstructions corresponding to localized particles (subtomograms) or the projection data underlying them (pseudo-subtomograms) can be extracted and precisely registered (alignment) prior to averaging, yielding an increasingly high resolution from in situ data in recent years [7,8,9].

For any molecular interpretation of cryo-tomograms, particles first need to be detected within a tomogram [10]. Particle localization is hampered by the low signal-to-noise ratio (SNR) of the data due to the low dose (shot noise), which can be aggravated by molecular crowding in cells that decreases the intrinsic contrast of the sample. Moreover, the registration (alignment) of tilt images may have errors that aggravate due to gradual alterations in the sample due to radiation. Lastly, tilt-series typically do not cover more than ±60°, instead of the ±90° that would be required for complete 3D reconstruction. The resulting ‘missing wedge’ of data in Fourier space causes the characteristic blurring of features along the beam direction in reconstruction [1]. Therefore, detection methods typically need to be tailored to cryo-ET data.

The baseline for detection in cryo-ET is template-matching (TM), where a reference structure (e.g., simulated model or experimental structure) is cross-correlated with the tomogram in different orientations [11]. Local normalization of the intensities within the volume covered by the template, as defined by a specified mask, increases detection performance and is efficiently computationally implemented using fast Fourier transforms (FFTs) in 2D [12] and 3D [10]. Furthermore, constraining these correlations to the common areas of the template and local tomogram has also proven beneficial [13]. Deep-learning (for example, convolutional neural networks) methods, which aim to detect particles using supervised segmentation, are becoming more prominent. These methods have been demonstrated to outperform TM in silico with ideal, annotated training data [14,15]. However, for application to experimental data, the availability of annotated training data remains an obstacle. At this point, TM remains the most widely used for particle detection in tomograms, and will remain invaluable to the further development of alternative approaches because it serves as a benchmark.

In TM, exhaustive translational sampling can be efficiently achieved through FFTs, while angular sampling requires the explicit sampling of distinct orientations. Despite its popularity in the computational mining of cryo-tomograms, the effect of angular sampling on performance has not been studied. In practice, the computation time typically dictates the angular sampling as the algorithm scales as $O(N^{3})$ with N∝1/∆α, i.e., proportional to the inverse of the angular difference between adjacent sampling points. Nevertheless, the GPU implementation of common image-processing tasks such as interpolation and FFTs enables vastly increased sampling compared to the common CPU-based implementations, which has been instrumental in single particle analysis [16,17]

Irrespective of the detection approach, its performance is influenced by tomogram quality, which, in turn, is strongly influenced by the sample itself. The local resolution of tomograms tends to be highest for biochemically isolated macromolecules, which can be embedded in thin ice layers of high contrast, allowing for resolutions up to (15 Å)^−1^, while ~(40 Å)^−1^ can be achieved for in situ data [18]. However, only a subset of tomograms of datasets exhibits these resolutions, as registration of the particle projections tends to lead to significant errors. The cryo-sample suffers from beam-induced deformations that cannot be adequately described by the alignment parameters of a rigid body [19]. The local compensation for such motions in the particle projections (pseudo-subtomograms) has been key to the considerable improvements achieved in subtomogram averages [7,8]. Expanding on this concept, tilt-series have sufficient information to fit complex alignment models (polynomials) to displacements observed in tilt-series, which can improve reconstruction quality [20,21,22].

Here, we integrate GPU-powered TM into a workflow for particle detection and subtomogram averaging (STA). The GPU implementation of TM drastically speeds up the orientational search. The increased sampling substantially improves sensitivity and specificity compared to the angular sampling that is commonly applied. Furthermore, we use this method to quantitatively examine the effect of tomogram alignment and its correction on particle localization. GPU TM is part of a new release of PyTOM [23] and is also available as a standalone module, which integrates with other software such as M 1.0.9 [8] and RELION 3.1.4 [7] for more refined STA workflows.

## 2. Results

### 2.1. PyTOM Integrates with Common Tomography Software in a Workflow

We integrated PyTOM with popular cryo-ET software such as M 1.0.9 and RELION 3.1.4 to make TM and other features accessible to processing pipelines (Figure 1). While PyTOM provides a full processing workflow (covered in a previous work [23]), we did not mirror powerful CTF correction and local tilt series alignment from other packages, but rather supported conversion the common STAR file format, which is accessible from the graphical user interface (GUI). Here, we focus on the integration of PyTOM TM with RELION and M [7,8].

To illustrate the workflow, we demonstrate it on a specific example: we selected a subset of 81 tilt-series from our recent study on rapidly isolated ER microsomes with attached ribosomes [3]. As the first step of the workflow, we preprocessed the tilt-series in Warp [24] for motion correction and CTF estimation, prior to exporting stacks for alignment and reconstruction in AreTomo [22]. We illustrate the result for a reconstruction of an exemplary ER microsome, which was only denoised for visualization in Figure 1. In the example, we computed downsampled tomograms for a voxel size of (13.8 Å)^3^. Then, the PyTOM GUI supports running TM on the folder containing the reconstructed tomograms. To generate the template, we modulated a cryo-EM map (EMD-2938) with a contrast transfer function (CTF) and filtered it to the first zero-crossing, which is (40 Å)^−1^ in the example dataset. During TM, the maximum local correlation coefficient (${\mathrm{LCC}}_{\max}$; see Equation (3) in the Materials and Methods for a mathematical definition) over the set of orientations is calculated at each position in the tomogram with a spherical mask (350 Å in diameter for a ribosome) [23]. We determined a cutoff for the ${\mathrm{LCC}}_{\max}$ in a single tomogram with an abundance of ribosomes to obtain a robust Gaussian fit (Figure 1). We applied the same cutoff to the candidate extraction of all the other tomograms because we expected the consistent ice layer thickness throughout the dataset to yield similar results [3]. After subtomogram reconstruction in Warp from PyTOM’s particle annotations, we aligned the subtomograms in RELION. Finally, high-resolution reconstruction was performed in M. A 3D visualization shows the relative distribution of the ribosomes on the ER-vesicles, showing the polysomal organization at the ER membrane.

### 2.2. GPU Acceleration Enables Enhanced Rotational Sampling in TM

TM can be implemented efficiently on GPUs because it does not require disk access after the initial loading, leading to efficient (continuous) GPU usage. For the interpolations used to realize the template rotations, we make use of texture memory caches on the GPU to exploit the spatial locality of voxels. Texture memory is efficient for images and volumes because interpolation involves pixels that are proximal in space but not necessarily in computer memory. We compared the runtime on CPU parallelized with 16 processes to the runtime on a single GPU; that is, common resources for cryo-ET workstations (Intel Xeon E5-2630 v4 processors compared to an Nvidia GTX 1080 Ti graphics card). The tomogram size of 460 × 460 × 250 voxels corresponds to typical, strongly downsampled tomograms and the angular increment of 13°, resulting in 7112 rotations. This is a common choice for CPU implementations of TM [3]. The runtime is 399 s on the GPU compared to 7199 s for the parallelized CPU execution. The ${\mathrm{LCC}}_{\max}$ results display minor differences due to the numerical differences of cubic spline interpolation on CPU and cubic b-spline interpolation on GPU (Appendix A). As in our previous CPU implementation, the GPU TM supports parallelization to split the search over multiple GPUs to allow for further speedup [23]. The running times on 4 GTX 1080 Ti’s are 107 s. (1.8 min), 661 s. (11.0 min), and 8119 s. (135.3 min) for a single tomogram when searching 7112 (13° increment), 45,123 (7° increment), and 553,680 (3° increment) rotations, respectively. Thus, the computation time linearly increases with the number of angles sampled (Appendix A). The achieved speed-up allows for increased angular sampling within a practically relevant computing time. Importantly, the GPU implementation maintains an acceptable computation time for routine use, even for 3° sampling.

### 2.3. Increased Angular Sampling Increases Detection Specificity

We further investigate the effect of the orientational search of the template on the correlation with particles in the tomogram, and the sensitivity and false discovery rate (FDR), as defined previously [25]. We defined the required angular sampling of a template from the Crowther criterion [26], which defines the minimum sampling at a given resolution:(1)$$\Delta \alpha =\frac{1}{r\cdot d}\;,$$
where d is the particle diameter in Å, r is the target resolution of the tomogram in ${Å}^{-1}$, and Δα is the angular increment in radians. From Equation (1), it follows that the required orientational search increases with the particle diameter (Figure 2A): at a voxel size of 1/(2r) (Nyquist sampling), a voxel localized at the maximum radius of d/2 from the center is approximately rotated to a neighboring voxel upon rotation by Δα. We estimated the angular sampling for our template with a low-pass at (40 Å)^−1^ and a ribosome of 300 Å diameter to be Δα = 0.13 radians, or 7.6°.

We compared TM with the ribosome reference (EMD-2938) on a tomogram from the ER microsome dataset for different angular increments (13°, 7°, and 3°), while extracting 1000 candidates from each angular sampling. The 13° increment was used in the previous analysis of this dataset using template-matching on CPUs [3]. We can observe an increase in the ${\mathrm{LCC}}_{\max}$ when sampling more orientations, which is most pronounced when progressing from the 13° to the 7° search, indicating a better agreement between the template and the target (Figure 2B). The increase was strongest for the 200 highest-ranking particles, likely because those are the true positives (TPs) in the dataset.

We then fitted a bimodal model of two Gaussians (one for the background noise and one for the particles) for each angular sampling to estimate the sensitivity and FDR for different ${\mathrm{LCC}}_{\max}$ cutoffs. For a given cutoff, the sensitivity estimates how many relevant items are retrieved from the full set of true positives, while the FDR estimates the fraction of false positives in the retrieved items. We used the sensitivity and FDR to plot a receiver operating characteristic (ROC) curve and calculated the point with the largest rectangle under the curve (RUC) to assess the classification quality. An RUC value of 1 indicates a perfect classifier, while a value of 0.25 indicates a performance that is equal to a random classifier. The RUC point also represents the correlation coefficient cutoff with the best trade-off between sensitivity and FDR. We can observe that the RUC most strongly improves (from 0.91 to 0.98) when increasing the angular sampling from 13° to 7°, leading to better separation of the Gaussian curves and an almost perfect classifier (Figure 2C). This improved classification performance is reflected in the increase in sensitivity (recall) from 0.91 to 0.99 (Table 1). Although the ROC curve remains almost identical for the 3° sampling (Figure 2C), the ${\mathrm{LCC}}_{\max}$ increases and the true positive population separates further from the background compared to the 7° sampling (Figure 2B,C), meaning that, in other tomograms, it might improve the classification. In conclusion, the ${\mathrm{LCC}}_{\max}$ and particle classification substantially improve upon an increase in angular sampling to the Crowther criterion, while oversampling has further benefits.

### 2.4. Local Tilt Series Alignment Increases Particle Detection Fidelity

Next, we aimed to investigate the influence of tilt-series alignment on particle detection. To this end, we compared TM on reconstructions with global and local tilt-series alignment, both determined by the software AreTomo 1.3.3 [22]. Global alignment refers to parameters that assume a rigid body motion of the sample (in plane rotation, x and y translations), while local alignment refers to the x and y shifts that are estimated in patches on top of the global parameters and can account for beam-induced motion. These locally measured shifts are interpolated using a distance-weighting method to correct local motion at each pixel.

We compared TM at a 7° rotational increment in tomograms reconstructed with global and local alignment for tilt-series with low, medium and high residuals for a fitted rigid-body motion to the gold fiducials, as reported by IMOD [27]. We assessed the TM quality by plotting the ${\mathrm{LCC}}_{\max}$ of the 300 highest-scoring particles (Figure 3A). An increased ${\mathrm{LCC}}_{\max}$ indicates better agreement between the template and target structure and can thus serve as a measure for the reconstruction quality. Locally aligned tomograms display, on average, improved ${\mathrm{LCC}}_{\max}$ values, and those from tilt-series with high fiducial residuals show a particularly marked improvement in the ${\mathrm{LCC}}_{\max}$, indicative of improved reconstruction quality. These tilt-series likely feature deformations that cannot be adequately described by a rigid body. The improved correlation with the template subsequently also led to improved classification (Appendix A). While reconstruction improvements are difficult to visually appreciate for most tomograms, the gold markers in the tomograms with high residuals show less distortion in the locally aligned reconstructions (Figure 3B). Even though these effects appear subtle, the template correlation is sensitive to these improvements.

### 2.5. Correlation Scores in Lamella Correlate with FIB Beam Damage

Due to the observed sensitivity of TM to tomogram alignment, we were interested in the effect of particle damage. FIB milling is known to induce sample damage close to the lamella surface. Ga^+^ ions have recently been shown to damage ribosome particles in a ~60 nm zone by TM in 2D [28], while plasma beams were also shown to damage ribosomes by analyzing the B-factors after STA [29]. We used the latter, in situ lamellae of HeLa cells thinned with an argon plasma FIB (EMPIAR-11306), to test for sample damage. We reconstructed the tomograms at 8× downsampling to a voxel size of (14.8 A)^3^. A visual inspection of the denoised volumes shows abundant ribosomes (Figure 4A). Compared to the ER microsomes dataset, TM with a 3° increment results in lower RUC values of 0.75, which is likely due to the large background noise from the ~300 nm ice layers (Appendix A and Table 2).

We extracted particles based on the RUC cutoff at a ${\mathrm{LCC}}_{\max}$ of 0.17, with an expected FDR of only 0.05. A 2D projection of the coordinates reveals ice layers that are slightly tilted along the *y*-axis and ${\mathrm{LCC}}_{\max}$ values that appear higher towards the center of the layer (Figure 4B). To better visualize the dependence of the ${\mathrm{LCC}}_{\max}$ on the z-position in the ice layer, we aligned them to the principal components of the particle coordinates. Projecting the scores on the *z*-axis shows that the highest ${\mathrm{LCC}}_{\max}$ values reside in the center of the ice layer (Figure 4C). There was a notable decrease in correlation in a ~50 nm zone towards the edge of the ice layers, indicating sample damage, which is consistent with the damage depth determined by B-factor analysis after STA [29]. Thus, TM is sensitive to FIB milling damage in tomograms and may be used to obtain an initial approximation of FIB milling sample quality in cryo-ET.

### 2.6. Integration with RELION and M for High-Resolution STA

We further analyzed the particles that were detected in locally corrected tomograms in combination with the 7° angular sampling from GPU TM by exporting them to Warp/RELION/M for high-resolution STA. We extracted 12,343 particles from our ER microsome dataset with a LCC_max_ threshold of 0.19 (Figure 2C), corresponding with an expected FDR of 0.02. Subtomograms and 3D-CTF volumes were reconstructed at 4× downsampling in Warp and imported into RELION for initial STA and 3D classification. The 3D autorefine program quickly hits the Nyquist resolution of (13.8 A)^−1^. We imported the aligned subtomograms to M for structure refinement, resulting in a reconstruction with a resolution of (7.2 Å)^−1^ (Figure 5A; orange FSC curve in Figure 5B) and local resolution of (6.2 Å)^−1^ (Figure 5C).

We then attempted to remove the remaining FPs from the dataset by 3D classification in RELION to test if this could improve the final reconstruction quality. By setting a low T-value (0.05), filtering the initial reference to 1/(500 Å), and specifying 100 classes for 3D classification, we obtained one pure class of ribosomes (as inspected by eye) at a fraction of 0.78 of all particles, as well as one class that contained mainly ribosomes but also ice contamination at a 0.19 fraction, and the remaining 98 classes, forming a fraction of 0.025, were likely FPs (Appendix A). We then exported the clean class containing 9665 subtomograms to M for further refinement. Even though the cleaned set has a lower FDR than the original refinement set, the reconstruction resolution reduces to (7.5 Å)^−1^ (Figure 5B). This slight decrease in resolution indicates that the removal of TPs (from class 2) has a larger negative impact on the resolution than the benefit of removing some FPs. Thus, the particles that were automatically classified based on the LCC_max_ values already have a sufficiently low FDR for high-resolution subtomogram averaging. For ribosomes, the workflow allows for the removal of FPs from data in an automated fashion and could obtain high-resolution subtomogram averages, largely without human intervention.

### 2.7. Particle Detection Is Sufficiently Sensitive to Study Molecular Sociology

We were interested whether our automated workflow has the sensitivity required to detect polysomal arrangements, a short-range order of ribosomes chained to the same mRNA strand [3,30]. The arrangements differ for cytosolic and membrane-bound ribosomes. Low-sensitivity particle detection results in missing links in polysomes, making it harder to detect their characteristic 3D arrangements. The neighbor distribution function is less sensitive to missing particles than polysome assignment. To this end, we calculated the relative positions of each particle’s closest neighbors and summed those coordinates onto a voxel space. This density indicates the probability of finding a neighbor at the specified position in the volume. Technically, this map can be calculated from the STAR files obtained in PyTOM, or the refined versions from RELION or M.

We already observed additional densities in the subtomogram average of the detected particles corresponding to polysome neighbors (Figure 6A). As in our previous work [3], we observed a neighbor distribution that is characteristic for membrane-bound polysomes (Figure 6B). The plot shows densities for both the trailing (i − 1) and leading (i + 1) neighbors, and a second ring is also visible, which corresponds to the (i − 2) and (i + 2) neighbors. Our automated workflow, applied to a subset of the original tilt-series, has sufficient sensitivity to study the molecular sociology of ribosomes.

## 3. Discussion

### 3.1. Extensive Rotation Sampling and Integration into Tomography Workflows

We investigated TM as a function of angular sampling, accelerated by a GPU port of TM in PyTOM. The increase in computational speed compared to CPU implementations allows for a dramatically expanded orientational search at relevant timescales on the order of 1 h. Using ribosomes as an example, we show that the enhanced angular sampling substantially improves detection performance. An angular increment of 12° is commonly used in many cryo-ET studies due to the time constraints of CPU implementations, which would be appropriate for the TM of ribosomes at a target resolution of (63 Å)^−1^. Here, we show that the ${\mathrm{LCC}}_{\max}$ performs significantly better as a classifier when the angular sampling is adjusted to the (40 Å)^−1^ target resolution (7°). Thus, the signal in the range of (63 Å)^−1^—(40 Å)^−1^ contributes to efficient detection by TM. Our results demonstrate that an appropriate orientational search is essential to high-performance detection with TM. This parameter, however, is often not taken into consideration when comparing TM with other detection approaches, and is sometimes not even reported.

The functionality of PyTOM, most notably TM, can easily be integrated with other tomogram-processing software. We demonstrate the integration with M and RELION, which provides a ~(7 Å)^−1^ resolution ribosome structure from 10,000 particles without user intervention. We envisage that the possibility of combining extensive angular sampling TM with powerful STA approaches will be highly beneficial to explore the structural biology of large macromolecules such as ribosomes in situ in a largely automated fashion [31,32,33,34].

### 3.2. Sensitivity of TM to Tomogram Quality

Triggered by our finding that the performance of TM in tomograms is sensitive to relatively high-resolution signals, we also investigated the dependence of TM on tilt series alignment. While the effects of local alignment correction for beam-induced sample deformation have been demonstrated qualitatively, e.g., in [21,22], here, we quantitatively show the improvements in tomogram quality through correlation with a template. Indeed, we observe a significant improvement in the ${\mathrm{LCC}}_{\max}$ as a classifier upon locally optimized reconstructions for those tomograms with high residual errors when aligned on a rigid body. The combination of extensive rotation sampling and improved tilt-series alignment is useful for automated high-sensitivity particle detection in tomograms.

Due to the sensitivity of the ${\mathrm{LCC}}_{\max}$ to the observed alignment of tilt-series, we also investigated whether correlation scores reflect damage to imaged macromolecules. To this end, we performed TM within tomograms of lamellae of whole cells, where plasma ablation can cause structural damage to the created surfaces. Indeed, the observed correlation values are lowered in a ~50 nm layer, which is consistent with earlier analyses based on the B-factors determined in subtomograms. Compared to gallium FIB-milling damage, as measured by the 2D TM signal-to-noise ratio (SNR), the drop-off in ${\mathrm{LCC}}_{\max}$ occurs in a similar range [28]. Thus, the TM scores provide information on the depth of damage prior to subtomogram analysis and are an alternative measure to B-factor approximation [29] and 2D SNR [28]. The TM implementation does not incorporate local variations in the CTF and is hence expected to be less accurate than B-factor approximation and 2D SNR. Nevertheless, the advantage is that the analysis can be performed early in standard tomogram analysis workflows.

### 3.3. TM with 3D CTF and Dose Weighting

We did not investigate the effects of 3D CTF and dose weighting on particle detection as we found this beyond the scope of this work, and instead convoluted the template volume with a single CTF function and low-pass filter to (40 Å)^−1^, and a binary missing wedge mask. Nevertheless, particles in a tilt-series experience different defocus values due to sample tilting and the height in the ice layer. Additionally, the dose accumulation incrementally damages particles throughout the detection process. Here, we show that, with appropriate angular sampling, TM is sensitive to information of at least up to (40 Å)^−1^, and therefore likely benefits from dose-weighting. Indeed, it has already been shown that proper modulation of the template with a per-particle, per-tilt 3D CTF and dose-weighting leads to better correlation with the template structure compared to the use of a binary mask [24]. We expect that implementing 3D CTF and dose-weighting in PyTOM will also lead to improved particle detection.

### 3.4. 2D vs. 3D TM

TM in 2D on in situ high-dose 2D projections has recently been proposed as an alternative to 3D TM on cryo-tomograms [35,36,37]. An obvious advantage of the 2D approach is speed: data acquisition is substantially faster as only a single acquisition is required per field of view compared to a tilt-series, and the computation time is substantially reduced because the third dimension is not sampled exhaustively. To approximate the z-location in 2D, different defocus values are scanned, e.g., in 200 Å steps, prior to refining these values locally. The 2D approach aims to make use of high-resolution information, requiring fine local angular sampling (e.g., ~2.5°) after coarser global searches (e.g., 12.5°). Although the TM approach presented here, which relies on exhaustive translational and angular sampling, could potentially be accelerated by a hierarchical orientation search, the global search in the thirrd spatial dimension will always be comparatively slow. Nevertheless, our GPU implementation and parallelization makes the time requirements comparable to the data acquisition time for a tilt-series, even at 3° increment angular sampling, which makes it suitable for incorporation into standard tomography processing workflows.

Two-dimensional TM has been reported to perform notably better than 3D TM when classifying TPs from FPs [36]. Our results suggest that the sensitivity in 3D strongly depends on the angular sampling that is used. In our example of ribosomes bound to ER vesicles, the sensitivity was strongly increased when increasing the angular sampling from a from a 13° to a 7° increment (Figure 2C). In the comparison of 2D and 3D TM, 3D matching was performed with an increment of 20° compared to 2.5° for 2D [36]. Thus, our results suggest that the observed improved 2D classification performance may, in part, be attributed to the finer angular sampling. Moreover, 2D-matching was performed on exposures that were acquired prior to a tilt series and, accordingly, beam damage primarily accumulated in the 3D tomograms. Considering the sensitivity of 3D TM to beam damage observed in this study, the subtle structural degradation of ribosomes in tomograms may also have contributed to differences in classifier performance. In summary, TM in 2D is substantially faster than that in 3D, but more extensive investigations will be required to conclusively compare the approaches, independent of such efficiency considerations. A quantitative comparison would require the same experimental setup and unified reporting measures. Importantly, our work shows that angular and translational sampling must be comparable in such studies. Ultimately, hierarchical sampling schemes and local CTF modeling, as elegantly used in 2D TM [35,36], will be important to the potential use 3D TM as a potent classifier in relevant timeframes.

### 3.5. Outlook in Relation to Deep-Learning-Based Particle Picking

Recent developments in particle detection and identification have used deep learning. These methods have shown to be highly effective when applied to synthetic data with a ground truth [14,15] and faster at the prediction stage [38,39]. When trained on properly annotated experimental data, they can automatically identify ribosomes in cellular tomograms with high sensitivity [38,40], similar to the 3D TM approach presented here. However, for their wider adaptation, more exhaustively annotated training datasets need to be made publicly available, especially those with a larger variety of particles and used over a range of experimental setups for better network generalization. Deep-metric-based approaches that map particles in pre-trained embedding manifolds are an alternative that may overcome the problem of a lack of annotated experimental data [39]. Embedding networks have the advantage that they can be effectively trained on synthetic data to learn an optimal feature representation to classify subtomograms; however, they have a poor performance on macromolecules with low abundance. To improve this, synthetic training data need to simulate accurate sample distributions, molecular crowding and cellular ultrastructure [41]. Considering these challenges, TM still provides a valuable baseline for particle detection in tomograms and has the benefit of being able to provide orientational priors for STA and be used to measure tomogram quality.

## 4. Materials and Methods

### 4.1. Preprocessing

We used a dataset of isolated ER-derived vesicles collected on the Thermo Fisher Scientific Arctica 200 kV microscope equipped with a Gatan imaging filter and K2 detector [3]. The dose per frame was $0.24\;e^{-1}/{Å}^{2}$, and 8 frames were acquired from ±51° with a 3° increment according to a dose-symmetrical acquisition scheme [42]. Motion correction of the tilt movies was performed in Warp (1.0.9) with a 1 × 1 × 8 grid, i.e., a single x and y shift were fit to each frame in the movie. The CTF was estimated on the sum of motion-corrected frames with a 512 pixel window in the (34.0 Å)^−1^—(10.0 Å)^−1^, for a 200 kV beam, amplitude contrast of 8%, 2.7 mm spherical aberration, and a search between 1 and 6 μm defocus. MRC stacks of tilt images (.st), together with files containing the tilt angles (.rawtlt), were exported from Warp for tilt-series alignment and reconstruction in AreTomo.

Tilt-series of plasma FIB tomograms were downloaded from EMPIAR-11306. On the advice of the authors, we used 9 representative tomograms from the dataset acquired at the following positions: 18, 70, 86, 123, 135, 142, 152, 166 [29]. The EER files were summed to 10 frames per tilt and the gain file was inverted with the RELION (3.1.4) command line tool relion_convert_to_tiff before motion correction and gain correction with MotionCor2 (1.5.0) [43] assuming a single x and y translation per frame. Images were combined to stacks (.st), together with angle list files (.rawtlt), for further processing.

### 4.2. Tomogram Alignment and Reconstruction

AreTomo (1.3.3) was used for alignment of tilt-series and tomogram reconstruction. For both global and local alignment, the alignment z-height (-AlignZ) was set to 1000 voxels (unbinned), and the tilt axis to −88.7° (-TiltAxis). No dark images were excluded (-DarkTol 0.01). Final volumes were reconstructed using weighted back-projection, 8× downsampling, 2000 voxels along the *z*-axis and stored in xyz fashion (-FlipVol 1), resulting in a voxel size of (13.8 Å)^3^ and a volume of 462 × 478 × 250 voxels. For local alignments, 5 patches were set along x and y (-Patch 5 5), resulting in 25 patches. The global alignment was also exported as an IMOD transformation file (.xf) for Warp import (-OutImod 1; as we did not exclude dark images, this option was sufficient to obtain the transformation file).

For the plasma-FIB dataset, the tomograms were reconstructed with similar parameters; notably, the tilt axis was set with a 84° prior, and tilt correction was set with a 10 degree prior (-TiltCor) due to the stage angle offset for lamellae data collection. The reconstruction z-height was set to 2400 voxels (unbinned) and the alignment z-height to 1500 voxels (unbinned). No patch alignment was estimated due to the lower signal in these tilt-series. Final reconstructions were also 8× downsampled, resulting in a voxel size of (14.8 Å)^3^ and volumes of 512 × 512 × 300 voxels.

To calculate fiducial residuals, the tilt series were aligned in IMOD (4.10.29) using gold beads as fiducial markers for a rigid-body model fit [27]. Tilt series with a relatively high, medium and low residual were selected from the dataset and used to compare the reconstruction quality of globally and locally aligned tilt series via TM.

### 4.3. Tomogram Denoising

Cryo-CARE (0.2.2) was used to denoise tomograms for visualization (but not for template-matching, as it might remove the signal) [44]. To obtain even and odd reconstructions from the tilt series, the recorded movie of each tilt was split into even and odd summed frames with Motioncor2 (1.5.0) [43]. The tilt-series alignment was first calculated on the full tilt series, and then applied to the even/odd stacks. Training patches were set as 72^3^ cubes, of which 1200 were extracted from each training tomogram. A total of 500 patches were used for normalization of the training data. Training hyperparameters were set to 100 epochs, with 200 steps per epoch, 16 batch size, a U-net kernel size of 3, U-net N depth of 3 and U-net N first of 16, and a learning rate of 0.0004. The network was trained on a subset of 5 tomograms, and then applied to the whole set. Networks were trained separately for ER microsomes and plasma-FIB lamellae.

### 4.4. GPU Template Matching

The mathematical definition for the local correlation coefficient (LCC) between a template T and a larger search volume V over a range of N 3D points at position x in V and rotation υ (defined by three Euler angles) for T was adapted from [12]:(2)$$\mathrm{LCC}\left(x,\;\upsilon \right)=\frac{1}{P}\sum_{i=(1,1,1)}^{N_{x}N_{y}N_{z}}\frac{\left(T_{i,\upsilon }*W_{i}-\bar{T}\right)M_{i,\;\upsilon }\left(V_{i+x}-\bar{V}\right)}{\sigma_{T}\sigma_{M_{\upsilon }V}(x)}\;,$$
where W is a point spread function weighting, which is convoluted with the template, $\bar{T}$ and $\bar{V}$ are the mean of the template and search volume, $T_{\upsilon }$ and $M_{\upsilon }$ are the template and mask rotated to υ, $\sigma_{T}$ is the standard deviation of the template, $\sigma_{M_{\upsilon }V}$ is the local standard deviation of V under M, and P is the sum of the values in the mask. For efficient implementation, we used the Fourier space definition given in [12]. The maximum value of the LCC at each position x in the search volume is given by:(3)$${\mathrm{LCC}}_{\max}\left(x\right)=max\left\{\mathrm{LCC}\left(x,\upsilon \right):\;\upsilon \in A\right\},$$
where A is the set of orientations that are search, i.e., a list of evenly distributed Euler angles for a given angular increment, as discussed in our previous work [23]. For example, for a 13° increment, we sampled a list of 7112 evenly distributed Euler angles, while for the 3° increment, we sampled 553,680 unique orientations of the template. In the software, precalculated lists are available for different angular increments. Parallelization was also discussed in our previous work [23]; however, it was extended to distribution over multiple GPUs, where each GPU is monitored by a single CPU process. The algorithm can either be parallelized by splitting the angular search or by splitting the tomogram into subvolumes. The former is computationally more efficient, but the latter can be used to fit the data in GPU memory. A flag can be passed to the TM program, indicating the use of a spherical mask; this speeds up computation, as $M_{\upsilon }$ is identical for each rotation υ, meaning that the computationally expensive $\sigma_{M_{\upsilon }V}$ also remains identical.

For the GPU implementation, rotations were all completed via the voltools module, which makes use of texture memory caches on the GPU for optimal speed. The default interpolation method was filtered cubic b-splines, as these have the highest accuracy for non-smooth surfaces. Additionally, CUDA FFT-plans, accessible through the CuPy (10.6.0) [45] API in Python (3.8.15), were used to efficiently apply FFTs to the rotated template during the local cross-correlation routine.

### 4.5. Particle Localization

A ribosome template was generated from a single-particle cryoEM reconstruction of the human 80S ribosome (EMD-2938) [46]. For ER-derived vesicles, the template was modulated with a CTF function at 3 µm defocus, 200 keV voltage, 2.7 mm C_s_, and 8% amplitude contrast. The CTF was set to 0 after the first zero-crossing and further multiplied with a low-pass filter-cutting at (40 Å)^−1^, which approximately coincided with the first zero-crossing of the data. All filtering was performed at a voxel size of (1.72 Å)^3^, before finally downsampling the template to (13.8 Å)^3^ voxels. The final box size of the template was 34^3^ voxels. All template modifications were performed using the create_template.py script, which is included in PyTOM. We generated a spherical mask with a diameter of 350 Å and a soft edge that encompassed the ribosome template via create_mask.py (alternatively, this can be carried out via the GUI). For the plasma-FIB dataset, the template was modulated with the appropriate CTF parameters, and the final voxel length was set to 1.85 Å × 8 = 14.8 Å.

TM on the ER-derived vesicles dataset was performed with missing wedge angles of 39° (the tilt-series were collected between ±51° degrees, i.e., the wedge is 90° − 51° = 39°). The angular search was run with either 12.85° (7112 rotations), 7° (45,123 rotations), or 3° (553,680 rotations) increments, as indicated by the text in figures. Jobs were run with volumes, split into four subvolumes by splitting both the *x*- and *y*-axis, and each subvolume was assigned a GPU (NVIDIA GTX 1080 Ti, Santa Clara, USA) for speed-up. For the plasma-FIB dataset, the used parameters were identical, except for the angular search, which was kept constant at 7°. The results included a score volume and an angle volume. The score volume contains the ${\mathrm{LCC}}_{\max}$ at each position in the tomogram, and the angle volume contains the corresponding index of the angle in the angular search list. For both datasets, 1000 initial candidates were extracted with a 9-voxel masking radius in order of decreasing ${\mathrm{LCC}}_{\max}$ values.

### 4.6. True Positive Estimation

True positive estimation was performed on the 1000 candidates by fitting a bimodal distribution to a histogram of the ${\mathrm{LCC}}_{\max}$ values (bin size 0.015). The underlying assumption was that the scores were from two Gaussian distributions, one for background and one for the true-positive particle population. To constrain the least-squares optimization, the noise distribution was initialized and bound by the mean and standard deviation of the score volume. The peak height was calculated as $N/(w\sigma \sqrt{2\;\pi )}$, where N was the number of voxels, w the histogram bin width, and σ the standard deviation of the score volume. From the bimodal distribution, the number of true positives (TP) in the particle population was estimated, as well as the sensitivity (or recall) given by $\frac{\mathrm{TP}}{\mathrm{TP}+\mathrm{FN}}$ and false-discovery rate (FDR) given by $\frac{\mathrm{FP}}{\mathrm{FP}+\mathrm{TP}}$ of the classification when sliding the classifier over different local correlation coefficient thresholds (where TP is true positives, FN false negatives, and FP false positives). The point on the curve that is the closest to the upper left corner, which forms the smallest rectangle with the corner (RUC), was used as the cutoff for the classifier, as it is commonly preferred and contains the best balance of TPs and FPs.

### 4.7. Particle List File Conversion

All particle annotations in PyTOM were stored in XML format and specify the origin tomogram, particle coordinates, and orientations. For integration with Warp and RELION, the annotations can be converted to the STAR format via the convert.py script. The coordinates in PyTOM specify xyz positions in the tomogram but can be converted to angstrom coordinates by providing parameters to the script. For Warp, this is not necessary, as the coordinate measure can be specified upon import. The PyTOM anti-clockwise ZXZ Euler angles are converted by transformation to rotation matrices and decomposition to ZYZ Euler angles, which are finally multiplied by −1 to obtain clockwise angles, as in RELION (same convention as in XMIPP, SPIDER and FREALIGN).

### 4.8. Subtomogram Averaging

Subtomograms were reconstructed in Warp (1.0.9) after importing AreTomo’s global alignment parameters (.xf file) and the particle annotations from PyTOM. To ensure the correct import of particle annotations, the tomogram dimensions in Warp were converted to the unbinned dimensions of AreTomo’s reconstructions. Subtomograms were first reconstructed at 4× downsampling, resulting in (6.9 Å)^3^ voxels. Subtomograms, CTF volumes, and corresponding STAR file were transferred to RELION (3.1.4) for STA and classification [47]. The subtomogram alignment was first optimized using RELION’s 3D auto-refine procedure, resulting in a (13.8 Å)^−1^ resolution reconstruction (~Nyquist). To remove FPs from the dataset, the resulting annotations were classified with 100 classes, the 3D auto-refine average filtered to (1/500 Å), and a T-factor of 0.1.

We refined two sets of particles in M: (i) the output from 3D auto-refinement (i.e., particles only classified by TM in PyTOM); (ii) the best class from the 3D classification in RELION that was a 0.78 fraction of set (i). Sets (i) and (ii) were then imported to M in separate projects for structure refinement with a mask around the full ribosome. Over multiple rounds of refinement, the following parameters were optimized: 2 dose-dependent sampling points for per-particle pose optimization, a 5 × 5 image warp grid, anisotropic magnification, and a 3 × 3 × 2 × 10 volume warp grid.

### 4.9. Neighbor Density Plotting

Subtomogram data can either be read from a PyTOM-style XML file or a STAR file from RELION or M to plot neighbor densities. For each ribosome, we took the vectors between itself and its n closest neighbors (n = 4), excluding neighbors closer than 100 Å (for ribosomes) to remove potential clashes. The vectors were rotated with the inverse orientation of the respective ribosome, resulting in the coordinates of neighbors in the coordinate system of the aligned reference. We fit a plane through the relative coordinates to estimate the orientation of the ER membrane, and then rotated all points to place the membrane along the xy-plane. These vectors were sampled on a 3D histogram, with voxels corresponding to (20 Å)^3^, and divided by the total number of analyzed neighbors to indicate the probability of finding a neighboring ribosome in each voxel. The histograms can be projected on the xy-, xz, and yz-planes to visualize the density of the neighbors surrounding ribosomes.

### 4.10. Particle Atlas Visualization

The 3D visualization of annotated particles in tomograms was performed with the ArtiaX plugin (0.3) [48] in ChimeraX (1.6.1) [49]. The particles were loaded from the STAR file produced by structure refinement in M and the final average was filtered to a resolution of (14 Å)^−1^ before plotting at the respective positions in the tomogram.

### 4.11. Code Availability

PyTOM is an open-source program available on GitHub (https://github.com/SBC-Utrecht/PyTom, accessed on 3 February 2023). We also provide a standalone module that only contains the GPU template-matching from PyTOM with only a command line interface (https://github.com/SBC-Utrecht/pytom-template-matching-gpu, accessed on 23 August 2023). A Python script for automated processing from tilt-series movies to cryo-CARE denoised tomograms is also available (https://github.com/SBC-Utrecht/cryocare-from-movies, accessed on 23 June 2023). The voltools module that we used for affine transformations of CuPy volumes is also available as a separate Python package (https://github.com/the-lay/voltools, accessed on 3 February 2023).

## 5. Conclusions

The sensitivity of TM in electron tomograms has often been underestimated due to insufficient angular sampling. The main reason for sampling below the Crowther criterion has been that the computation time for appropriate angular sampling would have been prohibitive using CPU implementations. We provide a GPU implementation in PyTOM that greatly speeds up TM, enabling the angular sampling required for the automated high-sensitivity detection of ribosomes in tomograms of ER-derived microsomes, which we use for assessment. To further illustrate the sensitivity, we analyzed the responsiveness of TM to local tomogram alignment and FIB milling damage. Interestingly, we found that visually non-obvious gains in tomogram quality arising from local tilt series alignment result in significant improvements in particle detection. Overall, we were able to transfer the automatically detected ribosomes to other software for direct high-resolution subtomogram averaging. To easily integrate this with other software, we also provide a standalone GPU template-matching module. We expect that the software will be widely integrated into different workflows for the fast and reliable automated processing of large macromolecular complexes in cryo-ET.

## Figures and Tables

**Figure 1 ijms-24-13375-f001:**
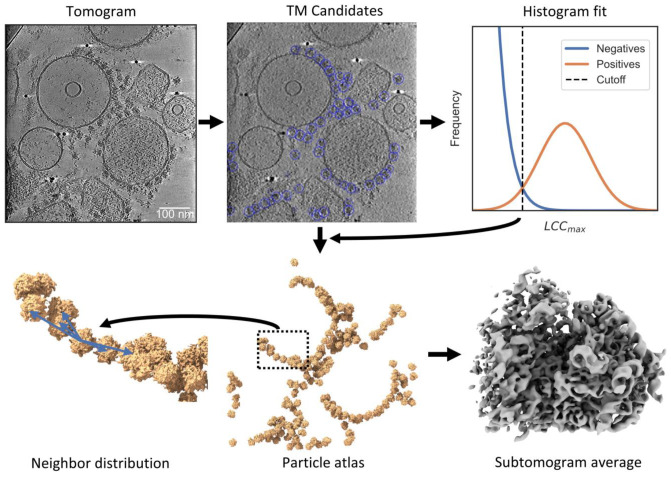
PyTOM’s particle detection can be integrated into a workflow for tilt-series processing. **Top left**: a tomogram, represented as a slice of a tomogram from the ER microsome dataset (denoised for visualization, displayed between ±2σ) is the input for TM in PyTOM. **Top center**: The particle candidates are indicated by blue circles in a slice of the tomogram. **Top right**: A model of two Gaussians is fitted to the LCC_max_ histogram to estimate the true positives in the dataset and to determine a cutoff to classify the candidates. **Bottom center**: These automatically annotated particles are shown as surface rendering in 3D. **Bottom right**: The extracted particles are the input for subtomogram averaging in M 1.0.9 and RELION 3.1.4. **Bottom left**: A zoom-in highlights the analysis of the spatial distribution of neighboring particles in PyTOM.

**Figure 2 ijms-24-13375-f002:**
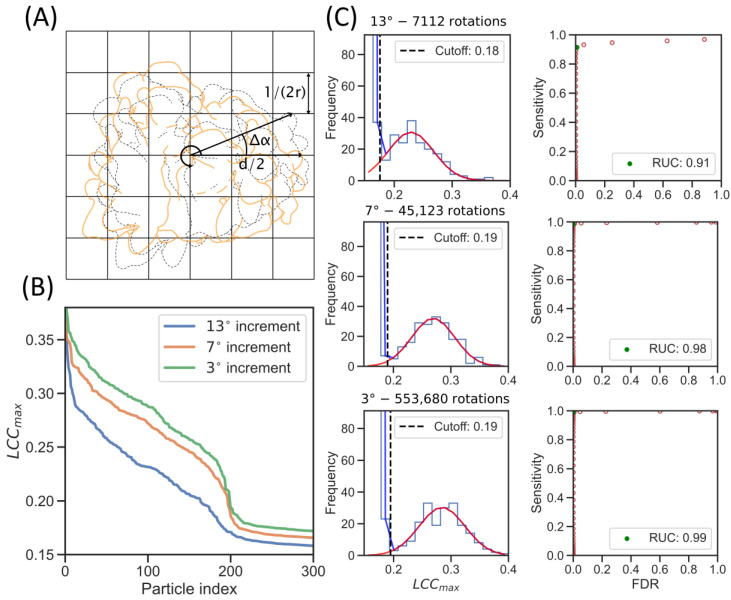
Increased rotational sampling improves particle detection. (**A**) Illustration of the relation between particle diameter and angular increment in Equation (1). The orange outline shows the original orientation of the template, while the dashed line shows the template after rotation. A point on the template is shown, which transforms by one pixel for the rotation Δα. (**B**) LCC_max_ values of the 300 highest scoring particles for rotation sampling at 13° (blue line), 7° (orange line) and 3° (green line). (**C**) Evaluation of performance of LCC_max_ as a classifier for different rotation samplings. Histogram of the LCC_max_ values on the left, with the Gaussian fitted to the true positives in the candidates (red line) and the bimodal model of the background and positives (blue line); the dashed line illustrates the LCC_max_ cutoff estimated from the RUC. On the right, the corresponding ROC curves are shown (red dots) with the RUC point (green dot) that provides a good balance between sensitivity and FDR.

**Figure 3 ijms-24-13375-f003:**
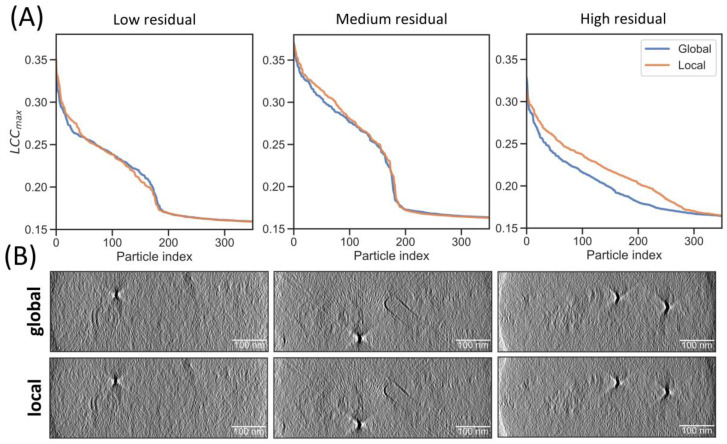
Local alignment improves particle detection in tilt-series with large deformation. (**A**) ${\mathrm{LCC}}_{\max}$ values from TM were ordered from high to low for globally (blue) and locally (orange) aligned tomograms. Left tomogram has residual error mean and standard deviations reported from the IMOD fiducial model as 0.780 and 0.547, middle tomogram has 1.210 and 0.638, and right tomogram has 2.206 and 1.459, indicating the decreasing fit of a rigid-body model from left to right. (**B**) Slice of the tomograms along the *y*-axis (or xz-plane) shown between ±3σ.

**Figure 4 ijms-24-13375-f004:**
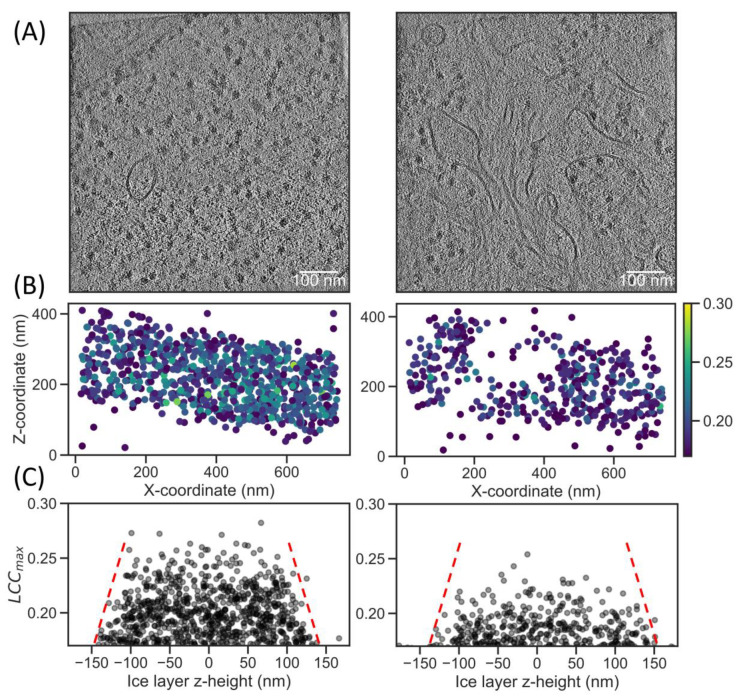
TM can indicate sample damage in plasma FIB milling. (**A**) Representative slice of tomograms from EMPIAR-11306 [29], denoised for visualization purposes and displayed between ±2σ (Position 106 and Position 142 of dataset). (**B**) A scatter plot of the coordinates projected along the *y*-axis and colored by their LCC_max_ values (blue = low, yellow = high). (**C**) Scores plotted by z-height in the ice layer; coordinates were first rotated to the plane with the least variation, then the median *z*-height was subtracted from all coordinates to set the center of the ice layer to z = 0. Red dashed lines were added as a visual aid to indicate the drop-off in LCC_max_ values when moving from the center to the edges of the ice layer.

**Figure 5 ijms-24-13375-f005:**
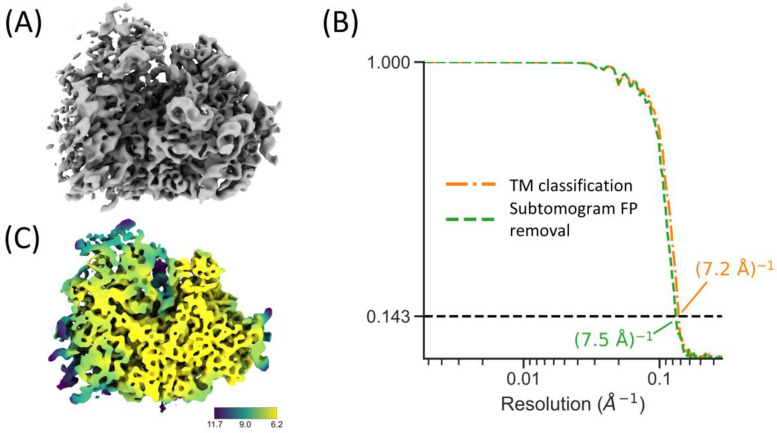
Detected ribosomes can be used for high-resolution averaging. (**A**) Refined ribosome from the 12,343 particles extracted from PyTOM. (**B**) FSC between half maps of the refined structure within a masked area for the set of particles automatically classified in PyTOM (orange) and after additional 3D classification of the subtomograms to remove false positives in the particle set (green). (**C**) Local resolution of the structure in (**A**); the values on the color bar are in Å units.

**Figure 6 ijms-24-13375-f006:**
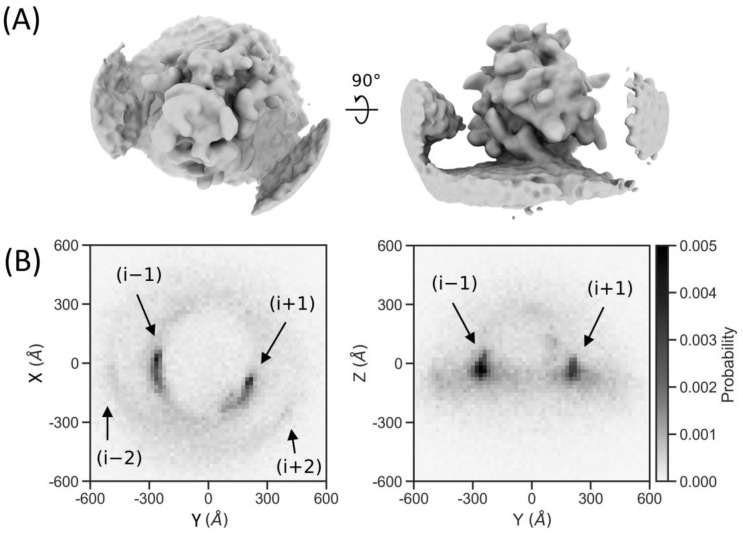
Ribosome neighbor density. (**A**) The subtomogram average of the particles filtered to (14 Å)^−1^ resolution shows a ribosome associated with the ER membrane, with two neighboring densities on the left and right, which are visible at lower density thresholds. (**B**) Neighbor density for the particles in the dataset calculated for 4 neighbors around each ribosome. The probability is projected onto the xy-plane (**left**) and yz-plane (**right**) corresponding with the view of the averages in (**A**), where the grayscales indicate the probability, with black corresponding to the highest probability of finding neighbors.

**Table 1 ijms-24-13375-t001:** Comparison of extraction threshold and sensitivity for TM with ribosomes on the ER microsomes. The median LCC_max_ refers to the median of the LCC_max_ values of the extracted particles above the determined cut-off. Sensitivity was estimated through the Gaussian curve fit.

Collection Parameters	Angular Increment (°)	Cut-Off (LCC_max_)	Median (LCC_max_)	Sensitivity
~160 nm ice layers, 200 keV—K2 Summit, ER microsomes of HEK cells	13	0.18	0.23	0.91
	7	0.19	0.27	0.99
	3	0.19	0.29	0.99

**Table 2 ijms-24-13375-t002:** Extraction threshold and sensitivity for TM with ribosomes on the plasma FIB-milled HeLa cells. The median LCC_max_ refers to the median of the LCC_max_ values of the extracted particles above the determined cut-off. Sensitivity (recall) was estimated through the Gaussian curve fit.

Collection Parameters	Angular Increment (°)	Cut-Off (LCC_max_)	Median (LCC_max_)	Sensitivity
~300 nm ice layers, 300 keV—Falcon 4, lamellae of HeLa cells	3	0.17	0.19	0.79

## Data Availability

The isolated ER-microsomes from HEK-cells dataset is available on request. The plasma-FIB milled lamellae of HeLa cells dataset is available online (EMPIAR-11306). All code is available via GitHub: PyTOM—https://github.com/SBC-Utrecht/PyTom, accessed on 3 February 2023; GPU template matching module—https://github.com/SBC-Utrecht/pytom-template-matching-gpu, accessed on 23 August 2023; preprocessing for cryoCARE—https://github.com/SBC-Utrecht/cryocare-from-movies, accessed on 23 June 2023; voltools—https://github.com/the-lay/voltools, accessed on 3 February 2023.

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
