# Peer review of "Extensive Angular Sampling Enables the Sensitive Localization of Macromolecules in Electron Tomograms"

_ijms, 2023, doi:10.3390/ijms241713375_

Round 1
Reviewer 1 Report
Chaillet reported an integrate GPU-powered TM into a workflow for particle detection and subtomogram averaging (STA) which allows users to comprehensive in situ structure analysis. It is an interesting manuscript. Some recommedations need to be considered further
1. A workflow may need to include
2. PyTOM’s particle, how it works and intergarated into the system that needs to be further explained
3. It would be good to have some data that were obtained from the new system in comparision with the other
Author Response
We thank the reviewer for going through the manuscript and providing helpful comments that led us to clarify the focus of the manuscript further.
- A workflow may need to include
We interpret this sentence as an advice that a full workflow description should be included and assumed the reviewer found section 2.1 not sufficiently clear in this regard. We now explicitly say that PyTOM covers a whole tomography workflow, but that we focus on the integration with software in the manuscript because they cover better-developed CTF correction and sub-tiltseries alignment for highest-resolution subtomogram averaging. This is why we do not provide a full PyTom workflow here. We clarified that we focus on the integration of GPU template matching in tomography processing, which allowed us to get important quanitication on the influence of angular sampling.
- PyTOM’s particle, how it works and intergarated into the system that needs to be further explained
We interpret this point as that it is not sufficiently clear how PyTOM’s particle detection works and integrates into the workflow. We refer to our answer to point (1), where we now made explicit that the whole workflow is not covered here.
- It would be good to have some data that were obtained from the new system in comparision with the other
We agree that a comparison with previous localization results was not entirely clear. We added a table to provide a quick overview of the angular sampling in relation to sensitivity and added a line in the 2nd paragraph of section 2.3 to emphasize how the dataset was previously analyzed: “The 13° increment was used in the previous analysis of this dataset [3].” In addition, supplementary figure 1 compares the CPU and GPU implementation and shows that results are consistent.
Reviewer 2 Report
The authors presented the paper "Extensive rotational sampling enables sensitive localization of macromolecules in electron tomograms" Thank you for an interesting work. I have only minor comments.
1) I highly recommend presenting the Conclusion section. The novelty and limitations of the work should be clearly mentioned in the Conclusion section and Abstract. The quantitative characteristics of the presented results should be clearly mentioned, too. It may highly improve the paper's significance for the readers.
2) Section 3.2. I recommend inserting some quantitative parameters of the data and discuss it with a comparison to the literature. Table with an essential parameters such as Sensitivity, LCC, SNR, etc. will be excellent.
3) Section 3.4. The same situation as for Section 3.2. Please, present any quantitative comparison with previously published works.
4) Section 3.5. Please, present any quantitative comparison. In such style of presentation it looks as a part of the review instead of experimental work or part of the Introduction section.
Minor
Fig. 1. Text is devided with a picture. (line 126)
Minor editing of English language required
Author Response
We thank the reviewer for carefully reading the manuscript and especially for the helpful suggestions in the discussion section.
1) I highly recommend presenting the Conclusion section. The novelty and limitations of the work should be clearly mentioned in the Conclusion section and Abstract. The quantitative characteristics of the presented results should be clearly mentioned, too. It may highly improve the paper's significance for the readers.
We thank the reviewer for the suggestion to present a Conclusion section, which is now added as section 5 (per the journals format). We emphasized the novelty and limitations of the work in the abstract and also made the quantitative character of the work more pronounced.
2) Section 3.2. I recommend inserting some quantitative parameters of the data and discuss it with a comparison to the literature. Table with an essential parameters such as Sensitivity, LCC, SNR, etc. will be excellent.
We added two tables, one in section 2.3 and one in 2.5 that highlight the characteristics of the datasets: mean ice thickness, electron acceleration voltage, and whether it is in situ or ex vivo. The tables also include the sensitivity of detection (as this is the most important parameter for template matching), and extraction parameters. We added a line in section 3.2 comparing the depth of damage of gallium and plasma FIB milling.
3) Section 3.4. The same situation as for Section 3.2. Please, present any quantitative comparison with previously published works.
A quantitative comparison with previous work here is difficult due to the difference in data collection approaches (as discussed in section 3.2). We attempted to compare with Lucas and Grigorieff (2023) as they also matched ribosomes in lamellae, which may appear similar to the dataset from Berger et al. (2023) that we analyzed. However, many experimental parameters are different: yeast cells versus HeLa cells, 200 nm versus 300 nm ice layer, gallium FIB milling versus plasma FIB milling, as well as the substantial variations in the individual FIB instruments and their contamination rates. Moreover, a sensitivity measure is not provided by Lucas and Grigorieff. Hence, we decided to refrain from a quantitative comparison and instead added the sentence to emphasize the difficulties in quantitative comparison: “In summary, TM in 2D is substantially faster than in 3D, but more extensive investigations will be required to conclusively compare the approaches independent of such efficiency considerations. A quantitative comparison would require the same experimental setup and unified reporting measures. Importantly, our work shows that angular and translational sampling must be comparable in such studies.”
4) Section 3.5. Please, present any quantitative comparison. In such style of presentation it looks as a part of the review instead of experimental work or part of the Introduction section.
We agree with the reviewer’s statement that this section does not give a direct comparison and updated the header accordingly. Although we do not give a direct comparison with these methods, we consider it important to acknowledge some of these development as they are relevant for the outlook of template matching. Importantly, we renamed the discussion section as an 'outlook'. Moreover, we did add a sentence to acknowledge that the method we present also works in an automated fashion similar to some deep-learning approaches: “When trained on properly annotated experimental data they can automatically identify ribosomes in cellular tomograms with high sensitivity [38, 40], similar to the 3D TM approach we present here.”
Minor
Fig. 1. Text is devided with a picture. (line 126)
Fixed.